# Online Extraction Followed by LC–MS/MS Analysis of Lipids in Natural Samples: A Proof-of-Concept Profiling Lecithin in Seeds

**DOI:** 10.3390/foods12020281

**Published:** 2023-01-07

**Authors:** João V. B. Borsatto, Edvaldo V. S. Maciel, Alejandro Cifuentes, Fernando M. Lanças

**Affiliations:** 1Laboratory of Chromatography, Institute of Chemistry at Sao Carlos, University of Sao Paulo, P.O. Box 780, Sao Carlos 13566590, Brazil; 2Laboratory of Foodomics, Institute of Food Science Research (CIAL, CSIC), Nicolás Cabrera 9, 28049 Madrid, Spain; 3Clemens Schöpf Institute, Department of Chemistry, Technical University of Darmstadt, 64287 Darmstadt, Germany

**Keywords:** online sample preparation, OLE–LC–MS/MS, lipids, lecithin, SiGO-C18ec column, seeds, lipidomics

## Abstract

Sample preparation is usually a complex and time-consuming procedure, which can directly affect the quality of the analysis. Recent efforts have been made to establish analytical methods involving minimal sample preparation, automatized and performed online with the analytical techniques. Online Extraction coupled with Liquid Chromatography–Mass Spectrometry (OLE–LC–MS) allows a fully connected extraction, separation, and analysis system. In this work, the lecithin profile was investigated in commercial sunflower, almonds, peanuts, and pistachio seeds to demonstrate that the concept of extraction, followed by the online analysis of the extract, could be applied to analyze this class of analytes in such complex solid matrices without a prior off-line solvent extraction step. The extraction phase gradient method was optimized. Two different analytical columns were explored, one being a conventional C18 (50 × 2.1 mm, 1.7 µm SPP) and the other a novel self-packed SIGO-C18ec (100 × 0.5, 5 µm FPP), which resulted in better separation. The analysis repeatability was investigated, and suggestions to improve it were pointed out. A characteristic ion with a m/z of 184, related to lysophosphatidylcholine structure, was used to identify the lecithin compounds. The temperature effect on the chromatograms was also explored. In short, it was found that the OLE–LC–MS approach is suitable for the analysis of lecithin compounds in seeds, being a promising alternative for lipidomics approaches in the near future.

## 1. Introduction

Sample preparation online coupled with the analytical instrument (usually LC–MS (Liquid Chromatography–Mass Spectrometry) or GC–MS (Gas Chromatography–Mass Spectrometry)) is a trend in modern analytical chemistry [1,2], particularly useful in the qualitative and quantitative analysis of multi-analytes in complex matrices, such as natural products. Automating sample preparation and joining it to analytical methods can reduce analytical errors, labor, and the amount of solvents utilized, creating simple and environmentally friendly approaches [3,4,5]. The sample preparation, alongside human error and equipment problems, are the three primary sources of errors in most analytical methods [6]. In-tube SPME is an example of a technique widely used for online coupling sample extraction with LC, which requires minimal sample preparation or even no sample pre-processing before starting the automated procedure [7,8,9,10]. Nevertheless, when the sample is a solid, in-tube SPME cannot be used without a previous sample preparation step to convert the solid sample into a solution compatible with the technique. Online extraction coupled with liquid chromatography (OLE–LC) is an exciting alternative for minimizing or suppressing this previous step, allowing the automated sample preparation of solid samples [11]. OLE–LC has been successfully applied to analyze natural and agro-products [12,13], soil samples [14,15], and other matrices [16,17,18]. To our knowledge, its applicability to the analysis of lipids in natural products has not been reported.

Lecithin is a general term given to amphiphilic lipid substances. Generally, it comprises a phospholipid connected to a glycerol group and different fatty acid groups; variations with groups such as glucose and cholesterol can also exist. In food industries, lecithin has been used with many purposes, such as an emulsifier, wetting agent, viscosity reduction, release agent, and in crystallization control [19,20]. In addition, it is a substance present in several natural sources [21], possessing a variety of fatty acids from source to source [22]. High-performance liquid chromatography coupled to MS has been an important tool for investigating polar and non-polar lipids [23,24]; supercritical fluid chromatography has also been applied [25] When an MS is unavailable, a derivatization process is often an option to analyze lipids by photodiode array (PDA) detection, owing to the lack of chromophores in the native lipids [26]. A review elaborated by Cajka and Fiehn summarizes the most relevant steps involved in lipid analysis. The report covered liquid chromatography, sample preparation, separation modes (such as reversed-phase LC, hydrophilic interaction chromatography, super-critical fluid chromatography, and two-dimensional liquid chromatography), mass spectrometry parameters, and data analysis [27]. Online approaches for the sample preparation of lipids have also been applied. For example, York and coworkers have developed a method to determine fat-soluble vitamins, mycotoxins, and hormones in hen egg yolk by sample cleanup by restricted access media coupled online to LC-MS [28]. Usually, lipids are analyzed in liquid chromatography (LC) with an octadecylsilane column (C18) or other more specialized columns [29,30,31]. The appropriate selection of a column stationary phase for lipids analysis is essential to avoid non-reversible binding between target compounds and the LC columns. Graphene oxide nanosheets covalently supported onto silica particles, chemically functionalized with octadecylsilica, and finally endcapped (SiGO-C18ec) is a promising solid material gaining new applications periodically. It has already been successfully applied as a sorbent for both extraction [32,33,34] and analytical columns [35]. Once OLE–LC can be considered a hybrid approach involving extraction and analytical steps, the SiGO-C18ec material can be considered worthy to be evaluated for this application.

This work aims to demonstrate that the OLE–LC–MS technique can be a promising tool for profiling lipid substances directly from a solid sample without non-automated and previous sample preparation. In addition, the effect of temperature on the analysis, repeatability, and column efficiency was also investigated.

## 2. Experimental

### 2.1. Chemicals, Samples, and Instrumentation

Chemicals: Chloroform (Avantor, Gliwice, Poland), formic acid (FA) (VWR International, Fontenay-sous-Bois, France), glass fiber filter 0.2 µm, lecithin refined standard 36,486 (ThermoFisher, GmbH, Karlsruhe, Germany), methanol (VWR international, Fontenay-sous-Bois, France), MilliQ water, line filter cartridge (Thermo scientific, Rockwood, USA), line filter hardware (Thermo scientific, Rockwood, NA, USA), 2-Propanol (VWR international, Fontenay-sous-Bois, France). Samples: almond seeds, pistachio seeds, peanut seeds, sunflower seeds.

LC Columns: C18 column of 50 × 2.1 mm and 1.7 µm superficially porous particles (SPP) (Phenomenex Inc., Torrance, NA, USA); SiGO-C18ec column prepared as described by Borsatto et al. [35], 100 × 0.5 mm and 5 µm fully porous particles (FPP) (details in Appendix A), and silica particles 50–60 µm mesh (Sigma-Aldrich Co St. Louis, MO, USA). Instrumentation: Accela LC system composed of an auto-sampler and a column temperature controller, a quaternary pump, and a PDA detector (Thermo Electronic Corporation, San Jose, CA, USA) joined to a TSQ quantum access triple quadrupole mass spectrometer (MS) fitted with an ESI source (Thermo Electronic Corporation, San Jose, CA, USA). Additionally, a cryogenic mill (Retsch, Haan, Germany) was used for grinding the seeds.

### 2.2. OLE System Assemble

First, the seeds are broken into pieces and mixed with silica in an Eppendorf in a ratio of 1:1 volume:volume. After that, it was macerated in a mill at room temperature until it formed a homogeneous-like powder with difficult differentiation between the silica particles and the seed material. Next, the lecithin refined standard powder was submitted to the same procedure before the analysis.

The OLE system assembly was: first, the line filter cartridge bottom (the part with the filter grade) was protected with a glass fiber filter of 0.2 µm. The filter was placed by pressing the upper part of the cartridge against the bottom part, cutting the filter in the middle. Next, the cartridge was opened again, and a small quantity of the mixture containing silica and the seed, just enough to partially cover the bottom of the cartridge (Figure 1A), was added. If an excess sample is added, the cartridge can be blocked. The addition of the sample to the cartridge is fast and takes less than one minute after the analyst acquires some practice. Next, the cartridge was placed in the line filter hardware in an inverse position and flushed with 100% water until the pressure stabilized (Figure 1B). After the pressure stabilization, the system was connected to the column inlet by the tubing, and the analysis started. For more details, see Appendix A. 

### 2.3. OLE–LC–ESI–MS Analysis

The extraction method, in which the mobile phase work in the extraction and elution process, employed a quaternary gradient using water, methanol, chloroform, and 2-propanol, as described in Table 1. It is important to run the entire method presented in Table 1 to minimize the carry-over effect once the proportion of the organic solvents in the mobile phase to extract some compounds from the matrix could be different from the proportion of the organic solvents in the mobile phase to elute them from the column, causing this compound to accumulate in the column. For the C18 column, the employed flow rate was 0.3 mL/min, and for the SiGO-C18ec column, the flow rate varied between 0.2 and 0.15 mL/min. The temperature was kept constant at 30 °C, except for the experiments evaluating the influence of temperature variation. The MS method used a positive ESI with a skimmer offset voltage of 20 V, a full-scan mass range between 50 and 1200 m/z, a scan time of 1 s, and a collision voltage of 10 V.

## 3. Results and Discussion

### 3.1. Column Selection

The first step of this study was the column selection. Two columns, a standard commercial C18 and a self-packed SiGO-C18ec, were compared. Figure 2 shows typical chromatograms obtained in both columns: SiGO-C18eccolumn in Figure 2A and C18 column in Figure 2B. The SiGO-C18ec (Figure 2A) column presented a different selectivity than the commercial C18 column. It is worth mentioning that this selectivity difference between SiGO-C18ec and C18 was already observed for small molecules [35]. Due to these observations, it was decided to use the SiGO-C18ec column in the OLE–LC–MS experimental series aiming to profile lecithin in seeds. However, the SiGO-C18ec column presented some limitations that must be highlighted. The small inner diameter (0.5 mm) results in higher column backpressure that required an appropriate pumping system capable of withstanding high pressures, making the mobile phase flow through the system successfully. Within such a context, this characteristic demands operating at lower flow rates. Furthermore, the SiGO-C18ec particle is a new phase for LC, and its characteristics are not yet fully known, making it hard to propose the dominant separation mechanisms participating in the retention of the compounds. However, possible interactions between the analytes and the different types of π-bonding existing in the graphene oxide network and with the hydrophobic C18 chain has to be considered. Owing to the endcapping procedure executed during the stationary phase preparation, hydrophilic interactions on the silanol groups can be excluded (at least to a relevant extension).

### 3.2. Repeatability

Repeatability is an important parameter to be evaluated when proposing an analytical method or a new application for an existing analytical technique for quantitative and qualitative analysis. Figure 3 shows two sets of comparisons. Figure 3A,B shows the comparison of the OLE–LC–MS separation of sunflower seeds in a C18 column, and Figure 3C,D show the comparison of the OLE–LC–MS separation of lecithin standard in a SiGO-C18ec column. Note that the signal intensity for both pairs of repetitions is similar, suggesting that the amount of sample placed in the holder was similar. However, the peak shapes in the pairs of chromatograms are slightly different, although it is possible to do a straightforward correlation between peaks inside the same pair of replicates. The main explanation for this observation is related to the eddy diffusion phenomena. When the sample is placed into the extraction cartridge, and the mobile phase flows through it, some voids could be formed, and the analytes disperse themselves in those regions. Once the formation of the voids is random and varies from sample to sample and cartridge to cartridge, it affects the repeatability. Once the target of this work was not quantitative, no attempt was made to improve the repeatability, but further investigation into the cartridge packing conditions would undoubtedly improve this factor. The literature shows the possibility of operating OLE–LC–MS quantitatively [36].

### 3.3. Temperature Selection

Temperature is an essential variable in the extraction of compounds from a complex matrix (e.g., natural seeds). To evaluate how the temperature was affecting the extraction performance in the OLE–LC–MS experiments, a C18 column was used once the SiGO-C18ec phase characteristics were not yet fully known. Therefore, the temperature variation could affect the separation mechanism of the SiGO-C18ec column and produce bias in the observation of the effect of the temperature in the OLE–LC–MS analysis. Figure 4 shows the OLE–LC–MS analysis at three different temperatures, 30, 40, and 50 °C. Minimal changes between the chromatogram profile and the MS intensity can be seen. So, it was decided to proceed with further experiments at 30 °C once the SiGO-C18 column had already been operating at this temperature.

### 3.4. Lecithin Profile in Seeds Samples

As a proof of the potential application of the OLE–LC–MS approach to the online extraction–separation analysis of lipids in natural products, four different seeds were used: almond, peanut, pistachio, and sunflower. Lecithins were identified based on the characteristic ion at m/z = 184, related to the lysophosphatidylcholine structure in the molecule [37]. Comparing the retention time and the MS spectra between the sample under study and the lecithin standard were also used to identify the investigated lecithin. Once a low-resolution triple quadrupole mass analyzer was used, confirming the identity of the fatty acids’ fragments composing each lecithin structure was pretty challenging. Figure 5 shows the chromatograms obtained from the lecithin standard, the samples, and the blank (OLE cartridge containing only the glass fiber filter and silica particles). Although the blank sample shows some peaks, the blank chromatogram presents lower intensity than the samples’ chromatograms. Consequently, it did not interfere in the observation of the peaks of the major compounds, but minor compounds in the sample might be suppressed. This can be confirmed by combining the MS spectra signals in this range and verifying that the characteristic lecithin MS spectrum could be visualized with no interference (the m/z signal recorded in the blank sample is displayed in the Appendix A). A possible explanation for this signal on the blank sample can be attributed to a potential carry-over effect provoked by the non-reversible bonding of some compounds from the samples to the column. This effect should be investigated in further work. However, when the retention time was higher than 18 min, the MS spectra observed became more complex, with several peaks presenting relative intensity closer to the base peak. When comparing the chromatograms in Figure 4, all samples presented a similar chromatogram, especially in the region having chloroform in the mobile phase (between 10 to 20 min).

Table 2 shows the retention time, the m/z values observed for each sample, and the lecithin standard when chloroform was present in the mobile phase. For the complete table with all lecithins observed in the lecithin standard, see the Appendix A. Seven lecithins were observed in the lecithin analytical standard, 2 in almond (A1 and A2), 2 in peanut (Pn1 and Pn2), 2 in pistachio (Pc1 and Pc2), and 3 in sunflower (S1, S2, and S3). Although the MS spectra observed were slightly different for each kind of seed, they presented a similar pattern compared to the lecithin standard and the characteristic ion m/z = 184, which confirms lecithin’s presence in the samples. Although the retention time and MS pattern presented similarities for all investigated samples, due to the complexity of the samples, it is challenging to confirm that they are the same substance. The sample identification step could be much improved by using a higher-resolution mass analyzer, including minor lecithins that could be co-eluted and were not investigated in this work. Despite these noticed drawbacks, the proposed approach demonstrates the possibility of automating a relevant step of the analysis of complex samples in difficult matrices, such as lipids in unprocessed seeds. This may provide an incentive for more studies expanding the focus on the qualitative analysis of other seeds or different solid complex matrices, using the fully automated online concept.

## 4. Conclusions

This work demonstrated that OLC–LC–MS is a technique that can be applied for online extraction–analysis of lecithin from seeds. Furthermore, it is a promising technique for performing direct lipids analysis in solid matrices. The SiGO-C18ec column was more suitable for this application in this approach than the conventional C18 column. Additionally, it is the first reported separation performed in the SiGO-C18ec column using chloroform and 2-propanol in the mobile phase, demonstrating that this column is an alternative when a less-polar mobile phase is required. Finally, although the repeatability obtained in this work was not as good as desired for quantitative analysis, the literature shows that this potential drawback can be overcome when desired. Shortly, the results obtained in this work prompted us to investigate the described approach further to perform lipidomic studies using this OLE–LC–MS approach.

## Figures and Tables

**Figure 1 foods-12-00281-f001:**
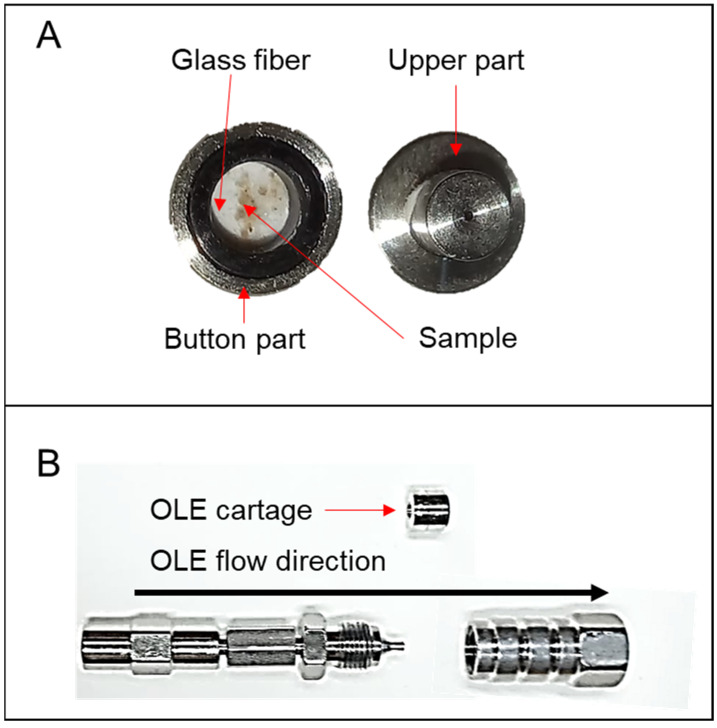
Set up of the analytical OLE system. (**A**) Example of OLE cartridge and the sample deposited on it. (**B**) Example of OLE hardware and flow direction during analysis.

**Figure 2 foods-12-00281-f002:**
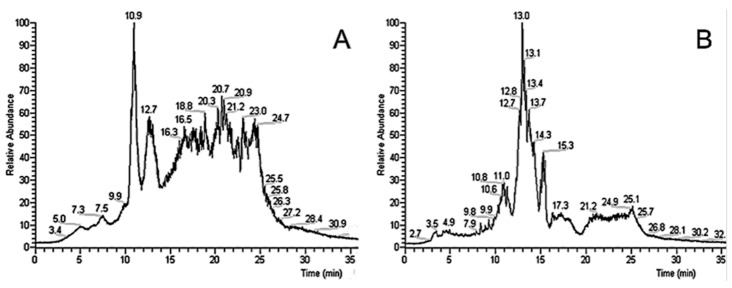
OLE–LC–MS analysis of sunflower seeds in quaternary gradient in (**A**) SiGO-C18ec (100 × 0.5, 5 µm FPP), at 0.2 mL/min and (**B**) C18 (50 × 2.1 mm, 1.7 µm SPP), at 0.3 mL/min. TIC chromatograms are in m/z range of 100 to 800. From 100% of H_2_O (0.1% FA) to 100% of methanol (0.1% FA) in 10 min, followed by a linear gradient to 90% of chloroform and 10% of methanol (0.1% FA) in 10 min, and the last stem of a linear gradient of 90% of 2-propanol and 10% of methanol (0.1% FA) in 10 min, followed by more 5 min of isocratic 90% of 2-propanol and 10% of methanol (0.1% FA) at 30 °C.

**Figure 3 foods-12-00281-f003:**
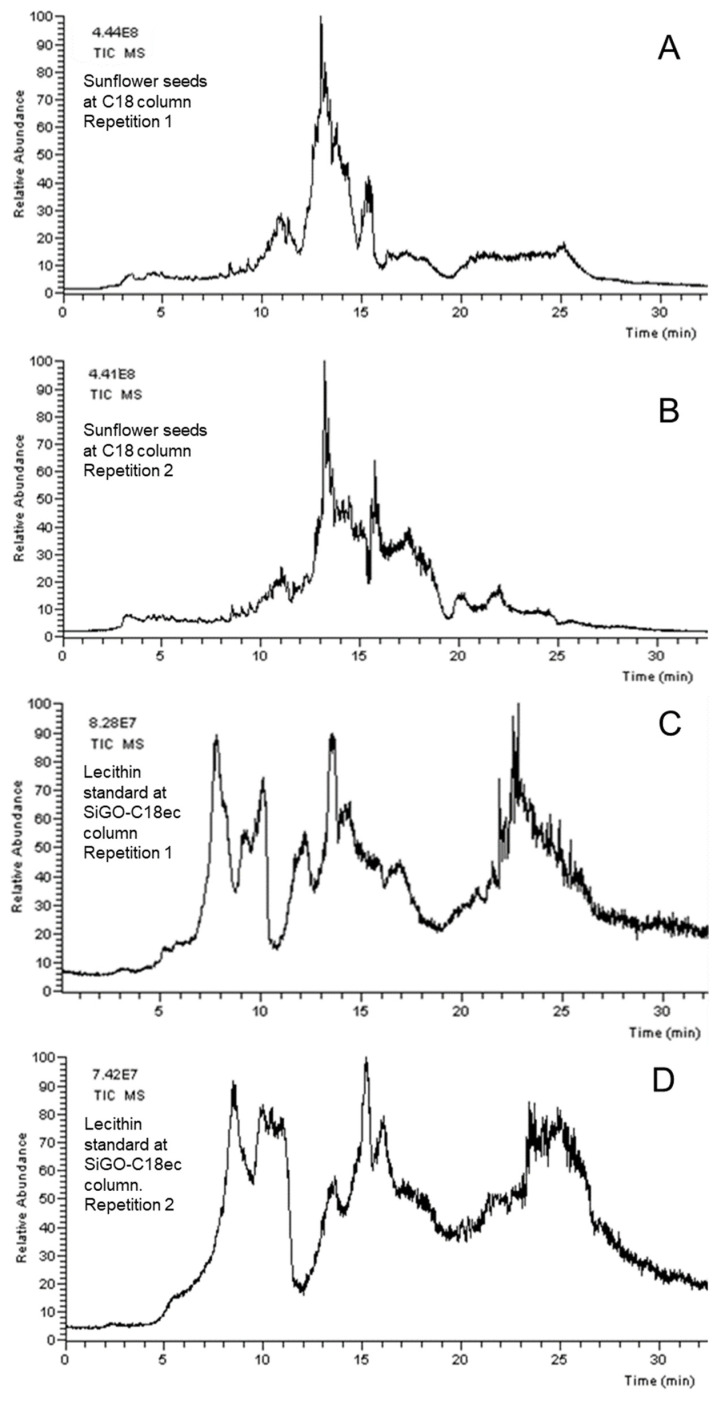
Repetitions of OLE–LC–MS analysis. Sunflower seeds in C18 column, 100 × 2.1, 1.9 SPP, at 0.3 mL/min (**A**,**B**) and lecithin standard in SiGO-C18ec, 100 × 0.5, 5 FPP, at 0.15 mL/min (**C**,**D**) in quaternary gradient going from 100% of H_2_O (0.1% FA) to 100% of methanol (0.1% FA) in 10 min, followed by a linear gradient to 90% of chloroform and 10% of methanol (0.1% FA) in 10 min, and the last stem of a linear gradient of 90% of 2-propanol and 10% of methanol (0.1% FA) in 10 min, followed for more 5 min of isocratic 90% of 2-propanol and 10% of methanol (0.1% FA) at 30 °C. TIC chromatograms in m/z range of 50 to 1200.

**Figure 4 foods-12-00281-f004:**
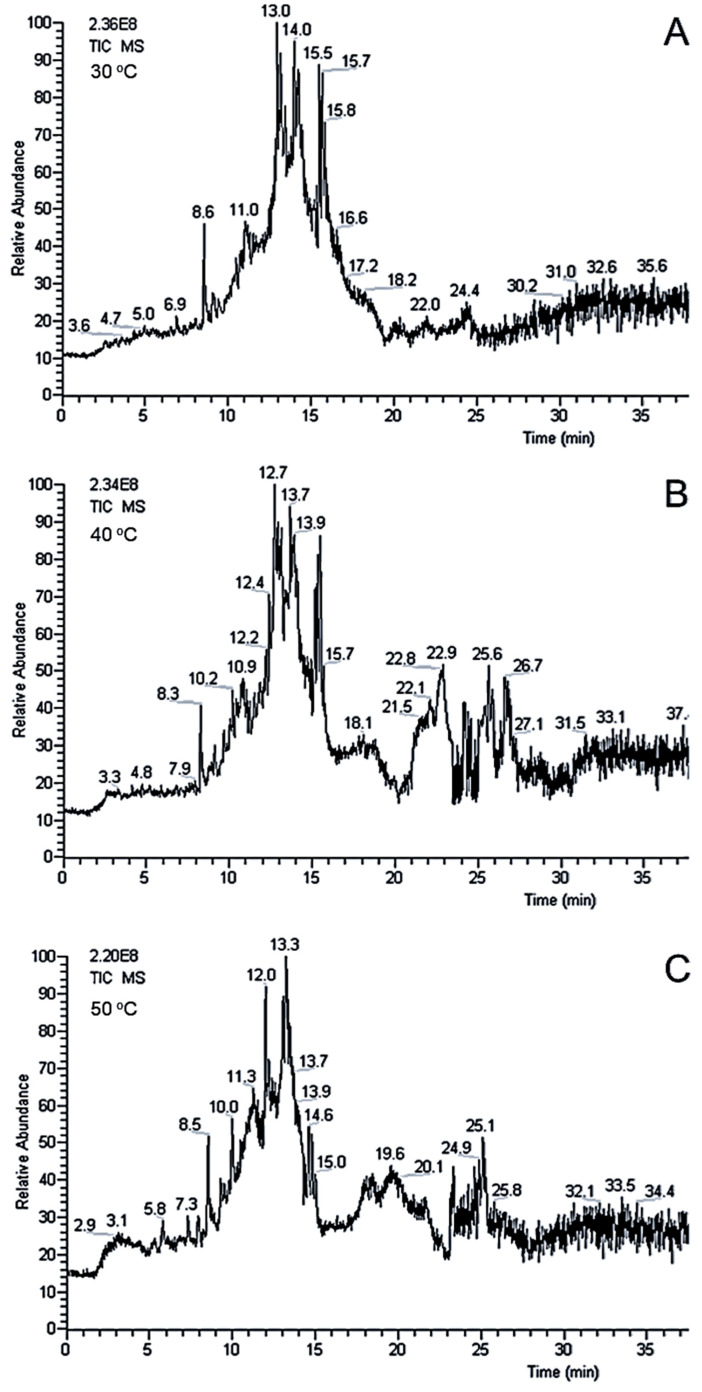
OLE–LC–MC analysis at (**A**) 30 °C, (**B**) 40 °C, and (**C**) 50 °C in the C18 column, 100 × 2.1, 1.9 SPP, at 0.3 mL/min. Quaternary gradient going from 100% of H_2_O (0.1% FA) to 100% of methanol (0.1% FA) in 10 min, followed by a linear gradient to 90% of chloroform and 10% of methanol (0.1% FA) in 10 min, and the last stem of a linear gradient of 90% of 2-propanol and 10% of methanol (0.1% FA) in 10 min, followed for more 5 min of isocratic 90% of 2-propanol and 10% of methanol (0.1% FA) at 30 °C. TIC chromatograms in m/z range of 50 to 1200.

**Figure 5 foods-12-00281-f005:**
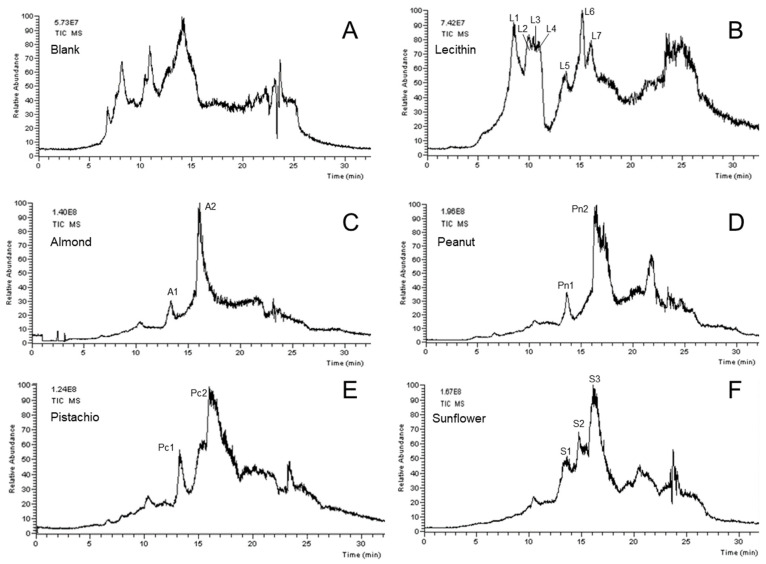
Comparison of the chromatograms obtained from (**A**) blank, (**B**) lecithin standard, (**C**) almond, (**D**) peanut, (**E**) pistachio, and (**F**) sunflower. Quaternary gradient going from 100% of H_2_O (0.1% FA) to 100% of methanol (0.1% FA) in 10 min, followed by a linear gradient to 90% of chloroform and 10% of methanol (0.1% FA) in 10 min, and the last stem of a linear gradient of 90% of 2-propanol and 10% of methanol (0.1% FA) in 10 min, followed for more 5 min of isocratic 90% of 2-propanol and 10% of methanol (0.1% FA) at 30 °C. SiGO-C18ec (100 × 0.5, 5 FPP), at 0.15 mL/min. TIC chromatograms in m/z range of 50 to 1200.

**Table 1 foods-12-00281-t001:** Time events of the quaternary gradient. The extraction and elution proceeded in a continuous flow.

Time	Water (0.1% FA) %	Methanol (0.1% FA) %	Chloroform %	2-Propanol %
0	100	0	0	0
10	0	100	0	0
20	0	10	90	0
30	0	10	0	90
35	0	10	0	90

**Table 2 foods-12-00281-t002:** Comparison of m/z values obtained in each retention time from a lecithin standard and from seed samples.

Lecithin Standard		Almond		Peanut		Pistachio		Sunflower	
tR	m/z (+)	tR	m/z (+)	tR	m/z (+)	tR	m/z (+)	tR	m/z (+)
13.6	184	13.3	184	13.3	184	13.3	184	13.5	184
			339		339				207
	575		577		577		575		575
	601		601		603		599		599
					643				
	758		758		760		758		758
	782		784		784		784		784
15.2	184					15	184	14.8	184
	306								207
							365		
									491
	575						575		575
	599						601		599
							617		615
	758						758		758
	782						784		782
	842								
									877
							883		
							919		
									935
16	184	16.1		16		16.3		16.2	184
	313								207
	575		577		577		577		575
	599								
			603		603		603		603
									615
	758								
	782								
			881		881		881		881
					882				
			905		905		905		904
					907				

## Data Availability

The data are available from the corresponding author.

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
