# Peer review of "Online Extraction Followed by LC–MS/MS Analysis of Lipids in Natural Samples: A Proof-of-Concept Profiling Lecithin in Seeds"

_foods, 2023, doi:10.3390/foods12020281_

Round 1

Reviewer 1 Report

This manuscript deals with the attempted development of automated online sample preparation in conjunction with the liquid chromatographic separation of commercial lecithin and seeds containing phospholipids. The manuscript deserves relevant improvements because is pervaded by a great deal of confusion and a lot of useless LC-MS chromatograms unable to persuade the readers of their content. While I cannot positively judge the difference between the two used reversed-phase columns, the manuscript does not offer a strong evaluation to distinguish the performances of the proposed online extraction (OLE) system and those related to the employed LC columns; one of these is a home-made prepared column, 100 x 0.5 mm and 5 um of fully porous particles compared with a second commercially available one of superficially porous particles (50 x 2.1 and 1.7 um). Unfortunately, no data are reported to distinguish the columns’ performance in quantitative terms because generic descriptions are given. The visual inspection of the reported chromatograms is not sufficiently valid and lends to unambiguous interpretations.

In the comments below I will try to explain my opinion and a few items, both of which will be listed roughly in the order of appearance, not necessarily reflecting an order of importance:

1.       Lecithins are mixtures of glycerophospholipids including phosphatidylcholines, phosphatidylethanolamines, etc. in which different FATTY ACYL GROUPS are involved.

2.      The use of alternative MS detection systems for intact phospholipids, such as photodiode arrays, is unfair also taking into account the need to carry out derivatization reactions.

3.      Although reversed-phase columns are described and used for PL separation in LC, not adequate mention is given to HILIC columns that greatly improved the separation of PL and should be referenced!

4.      Page 4. The authors stated that the two used columns exhibit “different selectivity”; such a sentence cannot be positively judged without showing single chromatographic peaks, using for instance specific PCs of the standard mixture. More details should be given to enforce the Author’s statement, which cannot be interpreted by TIC MS profiles.

5.       In Table 1, chloroform is proposed as a mobile phase solvent in a quaternary gradient without neither giving a reasonable explanation nor discussing all aspects of its use. How can such a solvent be compatible with polar ones such as water, methanol, and isopropanol being immiscible with them? How long does it take to mix methanol with chloroform (10:90) at 20 minutes? Please explain how chloroform can be mixed without trouble in the mixing chamber of the chromatographic pump in a continuous flow.

6.       Reading the manuscript, I thought that the primary purpose of the manuscript was the development of an online extraction system coupled with different RPCL columns; one of these columns was a homemade one not fully explored. The columns' features in the separation of phospholipids should be investigated before proposing a new OLE system. This is the most critical point of the manuscript that has to be taken into great consideration! This reviewer suggests focusing on a single column at a time because there is a great need to distinguish between variation due to sample extraction and column performances. The meaning of “repetition” in Figure 3 is not explained. As reported at point n.4, XIC chromatograms of distinguished PL (e.g. PC 18:0/18:1 or 18:1/18:2) should be used to evaluate the OLE system apart from the column performances. I firmly believe this manuscript deserves to be rethought and rewritten, clearly explaining the main aims.

Author Response

Dear reviewer,

The authors are thankful for your comments and observations. 

Please see the attachment for a point-by-point response

Kind regards,
João

Reviewer 2 Report

On-line extraction of lipids with LC-MS does not seem to be a common approach but the authors need to check this review article cited at the end of this review.

A recent related paper that could be discussed for comparison is also cited.

The  comparison of the standard C18 column with graphene-C18 column is interesting. 

Variability in the weighed out seed could likely be normalized for quantitative work.

Reproducibility of the extraction would seem to be dependent on how carefully the sample can be added to the extraction tube; what mass of powdered seed was used and  how difficult and time consuming is this addition?

The peak resolution in the chromatograms does not seem good enough for future potential quantitative work. Why is the response of the blank chromatogram so high in Figure 5A? Is this problematic for the MS identification?

How do these chromatograms compare to similar chromatography studies using off-line extraction? 

Was the quaternary gradient in Table 1 optimized?  If not, this needs to be done.

Comprehensive analysis of lipids in biological systems by liquid chromatography-mass spectrometry.T Cajka, O Fiehn - TrAC Trends in Analytical Chemistry, 2014 - Elsevier … primary focus of the study is to distinguish groups of samples. … 2D-LC of lipids can be performed in on-line or off-line modes… MS/MS data collection within a single LC-MS/MS analysis. …

On-line sample preparation for multiclass vitamin, hormone, and mycotoxin determination in chicken egg yolk using LC-MS/MS. JL York, RH Magnuson II, KA Schug - Food chemistry, 2020 - Elsevier

Author Response

(The authors gave the same response as above.)

Round 2

Reviewer 1 Report

The Authors state that the communication aims to present a “proof of concept” for an innovative online extraction method for lipids coupled to LC. The main problem is that without single chromatographic peaks, is not possible to evaluate the column’s performance, and this is not a matters of "proof of concept"!

Author Response

(The authors gave the same response as above.)

Reviewer 2 Report

The authors need to cite and describe in the Introduction the recommended references given in the first review (see below) so the context of the work is more clear. 

My other concerns were addressed.

Comprehensive analysis of lipids in biological systems by liquid chromatography-mass spectrometry.T Cajka, O Fiehn - TrAC Trends in Analytical Chemistry, 2014 - Elsevier … primary focus of the study is to distinguish groups of samples. … 2D-LC of lipids can be performed in online or off-line modes… MS/MS data collection within a single LC-MS/MS analysis. …

Online sample preparation for multiclass vitamin, hormone, and mycotoxin determination in chicken egg yolk using LC-MS/MS. JL York, RH Magnuson II, KA Schug - Food chemistry, 2020 - Elsevier

Author Response

(The authors gave the same response as above.)
